# Formulation Study on Edible Film from Waste Grape and Red Cabbage

**DOI:** 10.3390/foods12142804

**Published:** 2023-07-24

**Authors:** Olga Mileti, Noemi Baldino, Francesco Filice, Francesca R. Lupi, Maria Stefania Sinicropi, Domenico Gabriele

**Affiliations:** 1Department of Information, Modeling, Electronics and System Engineering (D.I.M.E.S.), University of Calabria, Via P. Bucci, Cubo 39C, I-87036 Arcavacata Rende, CS, Italy; o.mileti@dimes.unical.it (O.M.); francesco.fi@hotmail.it (F.F.); d.gabriele@unical.it (D.G.); 2Department of Pharmacy, Health and Nutritional Sciences, University of Calabria, Arcavacata di Rende, I-87036 Rende, CS, Italy; s.sinicropi@unical.it

**Keywords:** anthocyanins, grape marc, film viscosity, polyphenol, anthocyanin, purple cabbage

## Abstract

(1) Background: Recent research on the valorization of agro-industrial waste has attempted to obtain new products. Grape residue is a waste product used in the grape wine industry that is rich in anthocyanins, as well as leaves and waste parts from red cabbage processing. Anthocyanins, thanks to their various functionalities, can be recovered and used as active and intelligent agents in food packaging. Anthocyanins have antioxidant properties that help to prevent cardiovascular disease. (2) Methods: In this study, the process of extracting waste was studied using solvent and supercritical CO_2_ extraction. The obtained anthocyanins were used in starch-based food film formulations. Several formulations were studied using rheometric techniques and the effect of adding anthocyanins on optimal film formulation was investigated. (3) Results: Solvent extractions resulted in a maximum extraction yield. The extracts obtained were used for the preparation of coating and edible films, optimized in the formulation. (4) Conclusions: The addition of anthocyanins to films resulted in increased sample structuring and mechanical properties that are valid for applications, like dipping using coverage methods. The packaging is also attractive and pH-sensitive.

## 1. Introduction

Anthocyanins are compounds found in many foods, such as plant species. They can be extracted and obtained for utilization purposes as high-value compounds. Anthocyanins are natural plant pigments derived from the flavonoid family and their presence in food gives orange, red, purple, and blue colorations [1]. Anthocyanins are phenolic compounds and they derive from their respective aglycons (anthocyanidins), from which they differ following the addition of a glycosidic group (a sugar). The latter is composed of two aromatic rings linked to an oxygenated heterocyclic ring. Anthocyanins are anthocyanidins linked to sugar units [2,3]. Based on the number and position of hydroxyl groups and any sugars attached, as well as the number and type of aliphatic or aromatic acids based on sugars in the molecule, anthocyanins are distinguished into several forms [2,4]. Malvidin, pelargonidin, cyanidin, delphinidin, petunidin, and peonidin are six particularly common anthocyanins found in plants [1,3].

Anthocyanins are known in the literature for their antioxidant, anti-inflammatory, anticarcinogenic, and anticytotoxic properties [5,6]. Furthermore, they offer cardiovascular protection and have anti-diabetic and anti-obesity effects. In particular, the presence of anthocyanins in grapes and wine has been reported to lead to a beneficial association between these phenolic compounds and cardiovascular disease [7]. In fact, grapes are rich in anthocyanins, and about 75% of polyphenols are present in the seeds and skin; moreover, the possibility of recovering this raw material from the waste of the grape wine industry represents an interesting opportunity for waste valorization [8].

The leaves and waste parts of purple cabbage (*Brassica oleracea* L.) are also rich in anthocyanins that are widely available and easily extractable with water, but they are unlikely to be used much.

Anthocyanins also have an important characteristic in that they can change molecular structure depending on their pH; these structural changes lead to changes in color [1]. These variations are highly dependent on the matrix from which they are extracted. Their ability to vary in color with pH, from red in an acid condition to yellow in a basic condition, depends on the source from which they are extracted; thus, the pH dependence may differ [9]. Furthermore, their antioxidant capacity is proven to be influential on the production of active and intelligent packaging. In fact, the spoilage of some foods leads to the formation of compounds such as organic acids and volatile amines, resulting in pH changes [10,11]. Furthermore, their variability with pH can be used for the natural coloring of foods, making them naturally more appetizing and appropriate for children’s foods.

Film packaging prepared using anthocyanins have been used in the literature to monitor the spoilage of different food [11,12]. Different structuring agents can be used for film preparation, such as proteins and polysaccharides. Starch, in particular, is widely used because of its low cost, good mechanical properties, biodegradability, biocompatibility, and film-forming ability [13,14]. 

Extraction techniques involve the use of solvents, such as methanol, ethanol, acetone, water, or a mixture of these, under acidic ambient conditions to prevent the degradation of anthocyanins, but there has been a great focus on other efficient extraction techniques that are quick, eco-friendly, and qualitative [1,8,15]. 

In this study, anthocyanins were extracted from grape pomace using supercritical conditions with CO_2_ and from red cabbage waste using water. Following the extraction process under optimum conditions, anthocyanins were recovered and used to form starch-based packaging. The extraction from grape pomace was compared with the exhaustive classic solvent method. Furthermore, optimal conditions for film formation were studied, and the effect of adding the extracted anthocyanins was evaluated for better formulation. The correct formulation and dosage of glycerol and anthocyanins allow edible films to be obtained with a controlled consistency that can be used to protect foods and monitor their shelf-life, as well as being an additional source of natural antioxidants. The rheological characteristics of starting film-forming solutions, from which films and coatings were prepared, were evaluated to develop films, and, although there are several studies on this in the literature, the possibility of obtaining edible coatings with anthocyanins is poorly investigated in the literature, even if it is of great interest for the possibility of coloring or coating foods, making them more attractive to children. Therefore, the main purpose of this research is to investigate agro-industrial waste from cabbage and grape pomace that is still rich in antioxidant and active substances, and to attempt to use them in the formulation of edible coatings or films with active and intelligent characteristics.

## 2. Materials and Methods

### 2.1. Raw Materials

The grape pomace (GP) used in this study is a blend of grape pomaces (Gaglioppo, Cabernet and Merlot), provided by a local winery, with a moisture level of 47 ± 1% *w*/*w* and a water activity rate of 0.948 ± 0.001. The CO_2_ used for SFE extractions (purity > 99.99%) was supplied by SIAD Spa (Bergamo, Italy). For the SFE extractions, Terre di Ottawa and propylene wool (Applied Separation, Allentown, PA, USA) were used. The ethanol, methanol, acetone, and water used in this study were purchased from Sigma Aldrich Co. LLC (Italia) and the HCl 12N was purchased from Carlo Erba Reagenti (Italia). 

For film preparation, potato starch (MP Biomedicals LLC, Irvine, CA USA) and glycerol (Fisher Scientific Italia, Segrate (MI), Italy) were used in the formulations, as reported in Table 1, and distilled water was enriched with anthocyanins extracts.

### 2.2. Extraction Methods

The extraction of grape pomace was carried out in line with previous literature data [16] and extraction studies. Grape marc was extracted with methanol/12 N HCl (98:2 *v*/*v*), and a solvent-to-solid ratio of 1 mL solvent per g of extraction material and an extraction time of 24 h (4 times) were used during the conventional extraction experiment. 

A mixture of acidified methanol rich in anthocyanins was collected and a Rotavapor at 30 °C (Heidolph G3, Hei-VAP Value) was used to remove the solvent.

Supercritical CO_2_ extraction experiments were carried out in a laboratory-scale plant (Spe-ed SFE Applied Separations, Allentown, PA, USA) under 40 °C and 150 bar conditions, according to other literature studies [8,17] and a previous optimization. The GP for SFE extraction was pre-treated using N_2_ and crushed with a pestle to reduce its size. The crushed GP was sonicated at 35 kHz for 60 min using an ultrasonic bath Transsonic T310 (Elma Schmidbauer GmbH, Singen, Germany) at 25. Three extraction cycles of SFE were performed, and, for each extraction, 17 ± 1.5 g of GP was used, previously moistened with ethanol (according to the literature [17]) and mixed with Ottawa powders. The quantity of Ottawa powders used was fixed at a GP: Ottawa ratio of 50:30.

Each extraction cycle was performed for 1 h and the extractor worked discontinuously. Static phases were alternated to dynamic extraction phases, both lasting 30 min each. At the end of each operating cycle, the column of extraction was turned [18]. All samples were processed for three extraction cycles, as already mentioned. The extracts were collected in a volumetric flask and stored at −18 °C. In the end, the yield was calculated after weighing. Extractions were completed in triplicate and the yield maximum was reported using the following equation [18]:(1)$${Yield}_{max}\left(\%\right)=\frac{mass\;of\;anthocyanins\;[mg]}{mass\;of\;grape\;pomace\;feed\;[g]}\times 100$$

Extraction from red cabbage waste was carried out via the maceration of waste with distilled water at 70 °C for 15 min, in agreement with the literature [19].

All the extraction experiments were performed in triplicate. 

### 2.3. The Characterization of Extracts

The extracts obtained were analyzed via spectrophotometric measurements, using a UV-1601 spectrophotometer (Shimadzu, Duisburg, Germany), in line with the Folin–Ciocalteu method, to measure the concentration of total extracted polyphenols, evaluated as a gallic acid (GA) equivalent [20], and the total concentration of anthocyanins was evaluated, in agreement with the literature [19].

### 2.4. Film Preparation

The films were prepared with a preliminary film-forming solution (FFS), according to the procedure proposed by De Paola [21]. To be more specific, reagents were mixed with a magnetic stirrer (AREX Heating Magnetic Stirrer, Velp Scientifica, Usmate (MB) Italy) at 25 °C for 5 min; afterwards, the system was sonicated (Bandelin Sonorex, RH 102 K, Milano, Italy) for 10 min at 25 °C and 35 kHz. The obtained suspensions were heated to 80 °C for 45 min to allow starch gelatinization, while mixing was carried out with an overhead stirrer (RW20, IKA, Staufen, Germany) at 150 rpm. 

The FFS was used to form related films using the tape-casting method, according to previous studies [21], at 40 °C using a knife (SAFA-209/3, SAMA Italia s.r.l, Viareggio, Italy). A Mylar support was used for casting purposes to favor the removal step. The casted film was then dried for about 3 h at 65 °C in a forced convection oven (FD 53, Binder, Tuttlingen, Germany). Once cooled, the film was gently removed from the surface.

The anthocyanin solutions for the FFS were prepared from the extracts of cabbage and grape marc at an equal concentration of 3.6 ± 0.2 mg _anthocyanins_/mL. For the cabbage, only one extraction cycle was used, while for grape pomace, multiple extracts were collected to arrive at the same concentration. The FFSs for film packaging were prepared using the composition reported in Table 1.

Simulated edible coatings were prepared with two dipping steps, using the film-forming solutions with 24 h of aging. Inert polystyrene samples were immersed in these solutions at 40 °C for 10 s. Subsequently, the samples were dried in an oven for 2 h at 40 °C. The edible coatings were obtained at three different pH conditions (pH = 3.8; pH = 6.1; pH = 8.3).

### 2.5. Rheological Characterization

The FFS was characterized using steady-step temperature ramp test (SSRTRT) measurements before being heated at 80 °C in a temperature range between 25 and 90 °C, using a heating ramp of 2 °C/min and a constant shear rate of 0.1 s^−1^. After the gelatinization process, the FFSs were characterized by a frequency sweep test at 40 °C in the linear region to investigate the material microstructure. Furthermore, flow curves at 0.1 s^−1^ of the shear rate were performed to evaluate the suspension behavior. All the rheological tests were carried out using a rotational rheometer (HAAKE MARS III, Thermo Fisher Scientific, Braunschweig, Germany), a parallel plate geometry (ϕ = 50 mm, gap = 1.2 ± 0.1 mm), and a Peltier system to control the temperature. All measurements were conducted twice and, to reduce water evaporation phenomena during tests, silicone oil of 20 cSt (VWR Chemicals, Briare, France) was used.

The frequency sweep tests were interpreted using the weak gel model [22]: (2)$$G^{*}=A\omega^{1/z}$$
where *G** is the complex modulus, ω is the frequency, *A* is a measure of gel strength, and *z* is a measure of gel structuration.

The flow curves were interpreted using a power law:(3)$$\eta =k{\dot{\gamma }}^{n-1}$$
where *η* is the viscosity, $\dot{\gamma }$ is the shear rate, *k* is the consistency index, and *n* is the flow index.

The films were tested in elongational kinematics using the tensile machine Proline Z005 TN (Zwick Roell, Ulm, Germany). The film strips (size: 100 ± 5 × 20 ± 1 mm) were clamped at their ends by means of the pneumatic clamps (8195.01 of 20 N) of the tensile machine, starting from an initial distance of 50 mm and using a testing speed of 50 mm/min, according to other literature studies [14,23,24,25]. The elastic modulus was evaluated as the slope of the linear range of the stress–strain curve [26] and the elongation at break (EAB%), as follows [27]:(4)$$EAB\%=\frac{∆L}{L_{0}}\times 100$$
where ∆L and $L_{0}$ denote the variation in the elongation and the initial length of the film sample, respectively.

### 2.6. Scanning Electron Microscopy (SEM)

The surface morphology of the film was observed using a scanning electron microscope (FlexSEM 1000 II, HITACHI, Tokyo, Japan). The samples were allocated on double-sided adhesive carbon tabs, positioned on an aluminum sample stub. All pictures were acquired in low-vacuum conditions (50 Pa) using the BSE-COMP signal, with an accelerating voltage of 15 kV and a working distance of 6.5–6.9 mm [28]. Morphological analysis was carried out at 400× magnifications. 

### 2.7. Color Measurements

Color measurements were performed to evaluate the coloring effect of anthocyanins on the film using a colorimeter (Croma Meter CR-400, Konica Minolta, Tokyo, Japan), and the measurements were analyzed in the CieLab space. The colors were defined by color coordinates in CieLab Space: L*, the axis of lightness; a*, the axis of the red–green transition; and b*, the axis of the yellow–blue transition. Color changes were measured for samples with anthocyanins using Equation (5).
(5)$$∆E={\left[{\left(L_{i}^{*}-L_{0}^{*}\right)}^{2}+{\left(a_{i}^{*}-a_{0}^{*}\right)}^{2}+{\left(b_{i}^{*}-b_{0}^{*}\right)}^{2}\right]}^{0.5}$$

Preliminary calibration was then completed.

## 3. Results

### 3.1. The Extraction of Total Phenols and Anthocyanins

Red cabbage waste extraction offers solutions with an intense purple coloration, and, to assess its anthocyanin content, a spectrophotometric method was used, as reported in the Materials and Methods section. The quantity of total polyphenol was 1010 ± 20 mg GA/g_waste_, while the concentration of anthocyanins was found to be 17.7 ± 0.9 mg/g_waste_. The values obtained are in line with what has been reported in the literature for similar extractions [19]. The same procedure for the determination of polyphenol and anthocyanins in red cabbage was used for the GP extract, revealing concentrations of 18.2 ± 0.8 mg GA/g_GP_ and 1.2 ± 0.1 mg/g_GP_, respectively, according to previous literature studies [17].

### 3.2. Supercritical Extraction with Carbon Dioxide

From the GP, it was not possible to extract any antioxidants without the use of a pre-treatment. Therefore, optimization has led to the use of ethanol to favor the extraction of polar compounds, given the non-polar nature of CO_2_, as well as the use of the “cold technique” to reduce the size of the pomace. N_2_ crushing also increased yield as it promoted the formation of a porous structure, which in turn promoted mass transfer [17]. The quantity of total polyphenol was 3.42 ± 0.44 mg GA/g_GP_, while the concentration of anthocyanins was found to be 0.14 ± 0.06 mg/g_GP_. The values obtained are in line with what has been reported in the literature for similar extractions [17]. It is possible to observe that the solvent extraction results in the best type of extraction to recover the major quantity of antioxidants from GP, even if all the variables were optimized. This classic solvent extraction resulted in the best type of extraction to recover the major quantity of antioxidants from the GP.

### 3.3. The Rheological Characterization of the FFS

The FFS was prepared and characterized using rheological methods. In Figure 1, the rheological results are shown in terms of viscosity vs. temperature and viscosity vs. shear rate at 40 °C. It is possible to observe that increasing temperature increased the viscosity of all the samples because of the gelatinization reaction (Figure 1a). All the samples reached the maximum viscosity at about 70 °C and differences in the viscosity value were not appreciable.

The increasing glycerol percentage in the FFS did not affect the gelatinization temperature but only affected the final viscosity of the formed gels after the completed structuration, leading to the formation of less viscous films. This result is confirmed by flow curves (Figure 1b) which show a shear thinning behavior, typical for starch-based systems. Moreover, the viscosity values decreased as the amount of glycerol used increased, as suggested by other literature studies [21,29].

Frequency sweep tests were performed on the prepared gels in linearity, and the results are reported in terms of complex modulus (G*) shown in Figure 2, while the angle phase (δ) with a value of 25 ± 2 was not plotted because it is independent on both the glycerol content or the frequency. The gel consistency decreased as the amount of glycerol used increased, while the structuring of the material seemed to be scarcely affected by this, as confirmed by the phase angle values that are very similar to each other. All gels were found to have a solid-like behavior with phase angle values around 25°.

The frequency sweep tests were interpreted using the weak gel model, while the flow curves with a power law model and other parameters are reported in Table 2. By observing the parameters trend and evaluating A values, it is possible to point out that increasing the glycerol content decreased the gel strength because of a softening effect related to glycerol, while the network formation was only related to the quantity of starch, which did not change when constant.

Rheological characterization was also performed on the samples prepared using anthocyanin extract, and the data are reported in Figure 3. The viscosity vs. temperature trend on the FFS prepared with the addition of anthocyanins resulted in FFSs with higher viscosities than samples without anthocyanins after gelatinization (Figure 3a).

By observing the flow curves, it is possible to see a decrease in viscosity when the extract concentration increased. The decrease in viscosity is marked and the lower values are reached for the sample G1.5_100. In fact, on the G1.5_100 sample, it was not possible to perform oscillating analysis. The viscous behavior of the G1.5_100 sample can be attributed to an excess of anthocyanins, which interferes with the structure formation, leading to a highly unstructured sample [2,30]. This effect was particularly evident from A and Z values, which show an increase in both the strength and reticulation of gels with anthocyanin concentration. This behavior has also been found for gels prepared with low-amylose maize starch with an increase in persimmon tannin [31]. In agreement with our study, the starch used in this case also had a low amount of amylose to which polyphenols were added [31].

### 3.4. Film Characterization

Filmogenic solutions after drying were characterized by mechanical, color, and surface morphological analysis. Mechanical tests were performed using a tensile machine, and the results are reported in Table 3.

An increase in the glycerol percentage in FFS resulted in the formation of films with a lower elastic modulus and a greater ability to deform before rupture. This observed trend is in agreement with the literature results [32], as well as with the rheological analysis performed on the FFSs, from which it was observed that the addition of glycerol resulted in a less strong gel. For the G0.5 and G1 samples, during the elongation test, the maximum clump load was reached, and then it was not possible to calculate the EAB%. The breaking of these films was not investigated any further, as the aim of this study was to find suitable formulations for edible films and coatings with good plastic properties. From the rheological analysis, the best formulation was chosen for the addition of anthocyanins. The G1.5 sample was selected because of the high deformability appreciable from the value of the EAB%.

The addition of anthocyanins to the FFSs causes the formation of stronger films, according to other studies [12,33], as it is possible make observations based on an increment in the elastic modulus. The EAB% parameter tended to decrease with an increasing anthocyanin concentration, but the values were always very interesting if compared with other types of biodegradable films obtained with different formulations/biopolymers, reaching almost a double value compared to the literature data [12,33]. For other polysaccharide matrices, the same effects of an increased elastic modulus and an increased elongation at break were also found with the addition of anthocyanins [33].

The mechanical properties of films in elongational kinematic are in agreement with oscillating measurements on FFS, for which increases in structuration and strength were observed with an increase in anthocyanins [2].

The SEM investigation was performed in order to observe the morphology of the surface. The film is particularly sensitive to electron beams, so BSE-COMP analysis was used. The captured images are reported in Figure 4. Images of all samples prepared at varying glycerol contents were captured, but they are not shown as important surface microscopy differences were not detected. Therefore, the image of the sample without anthocyanins was chosen as a reference and was compared with samples with increasing concentrations of anthocyanins, as shown in Figure 4.

Microscopic surface inspection revealed important differences among samples with and without anthocyanins. In particular, it is observed that in the presence of anthocyanins, there is the formation of starch islands surrounded by spongy areas. This effect increases as the amount of anthocyanins loaded into the film-forming solution increases passing from a maximum, as it is possible to observe in Figure 4d which shows a less structured matrix. The images suggest the same result found by other authors for anthocyanin-loaded starch samples, where the detection is less defined because it is carried out in the secondary electron (SE) [34]. Mechanical experiments validated this microscopic result, and the effect of film strengthening followed by weakening in the presence of excess anthocyanins is confirmed in the literature [2].

From the color data reported in Table 4, it is possible to observe how the presence of anthocyanins significantly changed the color of the films, particularly by decreasing the brightness, as seen by the decrease in the L* parameter, but in particular by affecting the a* parameter, which was found to be more sensitive to red color variation. A less sensitive color parameter is b*.

The total color change expressed as ΔE was also affected by the addition of anthocyanins according to the amount used.

The G1.5_100 sample was found to be rich in anthocyanins, but it did not show an adequate structure because of its liquid consistency and low viscosity, making it impossible for it to be used it even for coating. The need to have a plastic and weakly textured gel is important because of its ability to both adhere to the substrate (which has to be coated) and to form a layer that does not slip off the substrate. For these reasons, in relation to the starch-glycerol formulations studied, the G2.5 sample prepared with anthocyanins solution 75% (G2.5_75) was chosen for coating formation purposes.

The coating was carried out at the natural pH of anthocyanin extraction (pH 6.1), resulting in a purple color. Using NaOH, the pH of the FFS was raised to a value of 8.3 and the color was blue. Using HCl, the FFS reached a pH value of 3.8, showing a pink color. Figure 5 illustrates the prepared samples.

## 4. Conclusions

Food waste from grape marc and red cabbage are valuable sources of polyphenols, and their recovery allows for their important development. In particular, the recovery of anthocyanins is interesting due to the properties of these substances, both in nutritional terms (by acting as antioxidants) and in terms of indicators of freshness (due to their ability to change color with pH). For this reason, the recovery of anthocyanins from grape marc and red cabbage waste was studied. Through extractive techniques, anthocyanin-rich extracts were obtained, and these were used for the preparation of films using tape-casting and for food coatings using a dipping method.

According to rheological measurements obtained, the G1.5 sample was identified as the optimal formulation for film formation and was used for the preparation of anthocyanin-enriched films.

Rheological measurements on the anthocyanin-enriched filmogenic solutions and the final films were in agreement and show that the addition of anthocyanins results in the formation of strong and consistent structures. However, the use of an excess of anthocyanins greatly weakened the structure. This rheological evaluation was also confirmed by SEM analysis. 

The G2.5_75 sample resulted in the best formulation to design an edible coating. The final coating color could be changed by varying the pH of the solutions. Furthermore, the coloration produced by using anthocyanins under different pH conditions also suggests their use for coloring foods by making them palatable for children consumption.

Finally, the use of anthocyanins extracted from food industry waste is an important way of enhancing the food supply chain. Their antioxidant effect and structuring capacity in starch-based packaging films make the anthocyanin–starch–glycerol system suitable for the preparation of active packaging films and intelligent packaging films.

## Figures and Tables

**Figure 1 foods-12-02804-f001:**
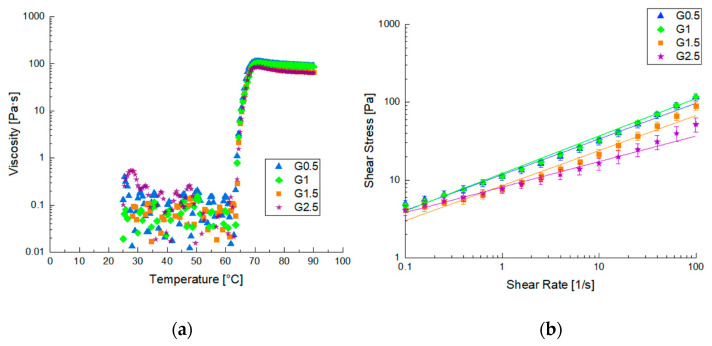
Diagrams of viscosity pre and after gelatinization process: (**a**) gelatinization diagram of different samples; (**b**) flow curves of starch-based gels at 40 °C.

**Figure 2 foods-12-02804-f002:**
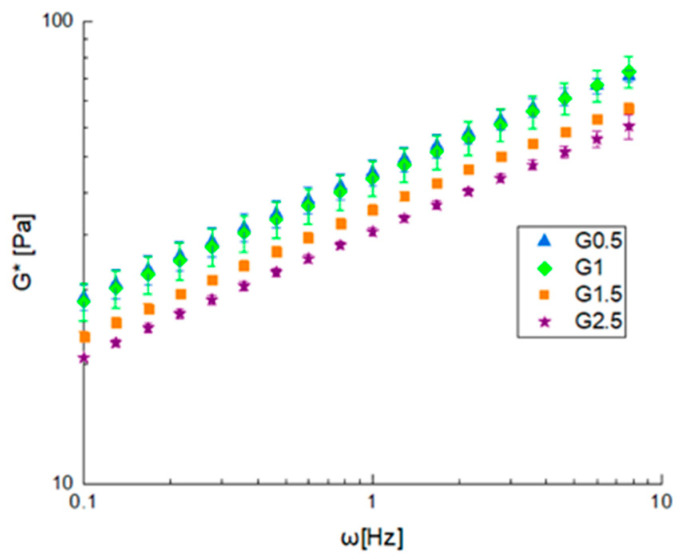
Frequency sweep test at 40 °C on gelled samples in terms of complex modulus (G*).

**Figure 3 foods-12-02804-f003:**
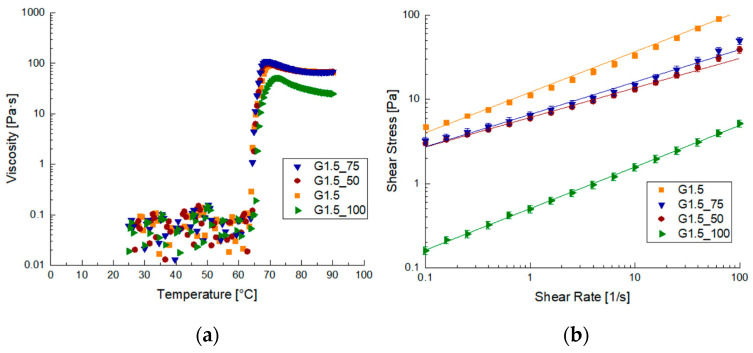
Diagrams of viscosity before and after the gelatinization process for samples with the addition of anthocyanins: (**a**) viscosity vs. temperature curves and (**b**) flow curves.

**Figure 4 foods-12-02804-f004:**
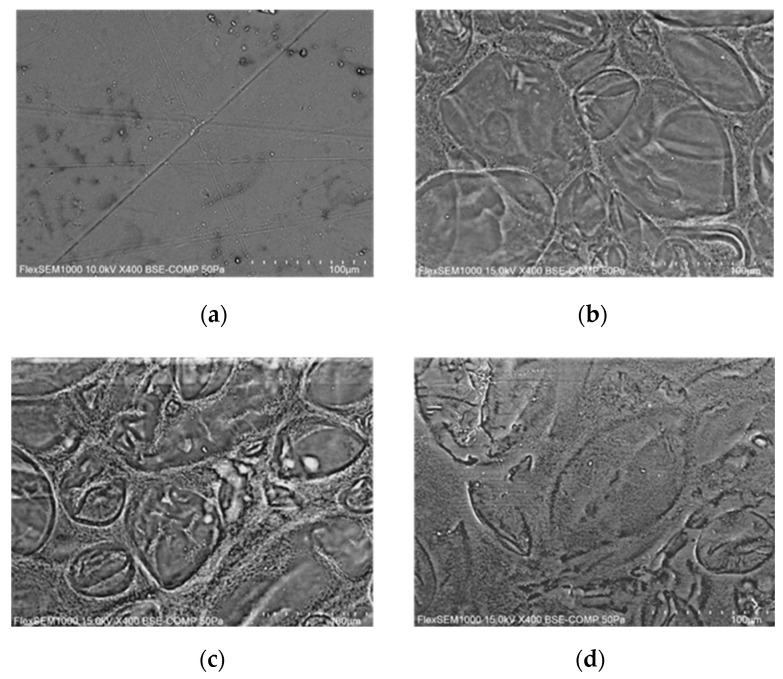
SEM images of G1.5 (**a**), G1.5_50 (**b**), G1.5_75 (**c**), and G1.5_100 (**d**).

**Figure 5 foods-12-02804-f005:**
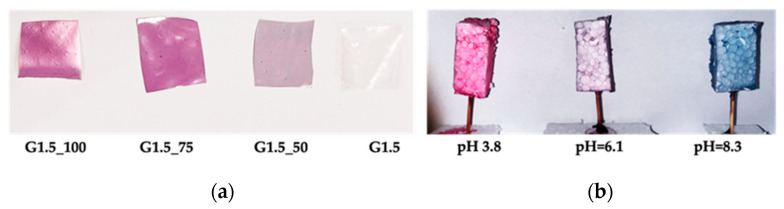
G1.5_100, G1.5_75, G1.5_50, and G1.5 samples after casting (**a**) and after coating at pH = 3.8, pH = 6.1, and pH = 8.3 (**b**).

**Table 1 foods-12-02804-t001:** The composition of film-forming solutions used for films.

ID Sample	Glycerol (g)	Starch (g)	Water (mL)	Anthocyanin Solution (mL)
G0.5	1	10	189	0
G1	2	10	188	0
G1.5	3	10	187	0
G2.5	5	10	185	0
G1.5_50	3	10	93.5	93.5
G1.5_75	3	10	46.75	140.25
G1.5_100	3	10	0	187

**Table 2 foods-12-02804-t002:** Data parameters of weak gel and power law models.

ID Sample	A, Pa*s^1/z^	z, -	k, Pa*s^n^	n, -
G2.5	34.9 ± 0.8	3.8 ± 0.1	7.90 ± 0.10	0.28 ± 0.01
G1.5	38.7 ± 0.6	3.9 ± 0.3	9.90 ± 1.00	0.34 ± 0.02
G1	45.6 ± 4	3.3 ± 0.1	7.60 ± 0.70	0.28 ± 0.02
G0.5	46.0 ± 3.0	4.1 ± 0.2	11.00 ± 1.00	0.34 ± 0.03
G1.5_50	68.6 ± 0.3	5.3 ± 0.1	6.10 ± 0.10	0.35 ± 0.01
G1.5_75	73.0 ± 3.0	5.0 ± 0.2	6.60 ± 0.20	0.38 ± 0.01
G1.5_100	-	-	0.50 ± 0.01	0.49 ± 0.01

**Table 3 foods-12-02804-t003:** Data parameters of the mechanical elongational test.

ID Sample	E, MPa	EAB%, -
G0.5	660 ± 40	-
G1	150 ± 30	-
G1.5	1.9 ± 0.3	154 ± 8
G2.5	1.1 ± 0.4	114 ± 1
G1.5_50	4.8 ± 0.9	146 ± 20
G1.5_75	7.3 ± 0.7	126 ± 21
G1.5_100	-	-

**Table 4 foods-12-02804-t004:** Data color parameters.

ID Sample	L*, -	a*, -	b*, -	ΔE, -	Thickness
G1.5	91.8 ± 0.1	3.24 ± 0.01	−4.42 ± 0.01	-	0.07 ± 0.01
G1.5_50	79.5 ± 0.7	6.8 ± 0.1	−7.4 ± 0.2	13.1 ± 0.8	0.06 ± 0.01
G1.5_75	75 ± 1	15.3 ± 0.8	−10.0 ± 0.3	21 ± 1	0.13 ± 0.01
G1.5_100	69.0 ± 0.7	19.8 ± 0.5	−10.4 ± 0.1	28.7 ± 0.9	0.16 ± 0.01

## Data Availability

The data used to support the findings of this study can be made available by the corresponding author upon request.

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
