# Peer review of "Formulation Study on Edible Film from Waste Grape and Red Cabbage"

_foods, 2023, doi:10.3390/foods12142804_

Round 1

Reviewer 1 Report

The main purpose of this work was to study the edible film from waste grape and red cabbage. The film properties such as rheological, color and surface morphology was evaluated.

In my opinion this article is well written, and it presents relevant results. Some specific comments are listed below:

Comment 1)

Title: Suggestion: “Formulation study of edible film from waste grape and red cabbage”. Since, the rheological study was not the only one to be approached in this research.

Comment 2)

The Graphical Abstract must be uploaded in a separate file and not inside the article.

Comment 3)

The Introduction contains updated references. However, in the final of introduction, it is usually presented the main objective of the work. Please insert it.

Comment 4)

The methodology is correctly referenced.

Comment 5)

The flow curve identifying as shear stress versus strain rate could be showed to demonstrate the pseudoplastic behavior of the samples, as well as the adjustment of the pseudoplastic model of Ostwald De Waele (equation 3). If it is possible, the authors could insert it?    

Comment 6)

The conclusions are consistent with the evidence and arguments presented in this work, but the following paragraph is not a conclusion:

“By rheological measurements on film-forming solutions, the flow and microstructural properties of starch-based gels were studied, and an evaluation of these properties was made at different amounts of glycerol used, identifying sample G1.5 as the optimal formulation for a film formation and this formulation has been used for the preparation of anthocyanin-enriched films. The anthocyanins were extracted by solvent and by SFE with CO2”.

I suggest change it for:

According to rheological measurements obtained the sample G1.5 was identifying as the optimal formulation for a film formation and has been used for the preparation of anthocyanin-enriched films.

Author Response

Reviewer 1

Comments and Suggestions for Authors

The main purpose of this work was to study the edible film from waste grape and red cabbage. The film properties such as rheological, color and surface morphology was evaluated.

In my opinion this article is well written, and it presents relevant results. Some specific comments are listed below:

Comment 1)

Title: Suggestion: “Formulation study of edible film from waste grape and red cabbage”. Since, the rheological study was not the only one to be approached in this research.

Thanks a lot to the reviewer, we rightly accepted the advice and changed the title.

Comment 2)

The Graphical Abstract must be uploaded in a separate file and not inside the article.

The graphical abstract has been removed and will be attached separately.

Comment 3)

The Introduction contains updated references. However, in the final of introduction, it is usually presented the main objective of the work. Please insert it.

Thank you for the advice, we have integrated the main purpose of the research into the final part.

Comment 4)

The methodology is correctly referenced.

Thank you to the reviewer

Comment 5)

The flow curve identifying as shear stress versus strain rate could be showed to demonstrate the pseudoplastic behavior of the samples, as well as the adjustment of the pseudoplastic model of Ostwald De Waele (equation 3). If it is possible, the authors could insert it?   

We have substituted the flow curves with the viscosity vs shear rate trend with the shear stress versus strain rate behaviour and the fitting curves.

Comment 6)

The conclusions are consistent with the evidence and arguments presented in this work, but the following paragraph is not a conclusion:

“By rheological measurements on film-forming solutions, the flow and microstructural properties of starch-based gels were studied, and an evaluation of these properties was made at different amounts of glycerol used, identifying sample G1.5 as the optimal formulation for a film formation and this formulation has been used for the preparation of anthocyanin-enriched films. The anthocyanins were extracted by solvent and by SFE with CO2”.

I suggest change it for:

According to rheological measurements obtained the sample G1.5 was identifying as the optimal formulation for a film formation and has been used for the preparation of anthocyanin-enriched films.

Thank you for the suggestion, we have amended the text.

Reviewer 2 Report

The manuscript deals with films containing anthocyanins. The anthocyanins were obtained by the authors using extraction from grape pomace and red cabbage. A number of physicochemical methods were applied for investigation of the films, film-forming solutions and starch-based gels with different amount of glycerol. The optimal ratio of components was selected in film-forming solutions to obtain the most suitable anthocyanin-enriched film with strong and consistent structure. The obtained results are interesting and have practical significance. At the same time, it is difficult to estimate the scientific novelty of the results. As it follows from the introductory section, anthocyanins have been already used for film packaging. It would be interesting to know the advantage of the investigated systems as compared with other ones and to compare the obtained experimental results with the literature data, maybe for other systems. Therefore, the manuscript is recommended for publication but only after the addition of this information, presumably in the sections of discussion and conclusions.

Minor remarks.

Please explain the abbreviation "GA", presumably on the line 132.

Some symbols are not defined in Eqs. (2) and (3).

Figure 2b does not contain any useful information and has to be excluded. It is sufficient to note that the phase angle does depend neither the glycerol content nor the frequency.

Author Response

Reviewer 2

The manuscript deals with films containing anthocyanins. The anthocyanins were obtained by the authors using extraction from grape pomace and red cabbage. A number of physicochemical methods were applied for investigation of the films, film-forming solutions and starch-based gels with different amount of glycerol. The optimal ratio of components was selected in film-forming solutions to obtain the most suitable anthocyanin-enriched film with strong and consistent structure. The obtained results are interesting and have practical significance. At the same time, it is difficult to estimate the scientific novelty of the results. As it follows from the introductory section, anthocyanins have been already used for film packaging. It would be interesting to know the advantage of the investigated systems as compared with other ones and to compare the obtained experimental results with the literature data, maybe for other systems. Therefore, the manuscript is recommended for publication but only after the addition of this information, presumably in the sections of discussion and conclusions.

Thanks to the reviewer for the suggestions. In accordance with what he indicated, we have briefly reported the advantages of the materials studied and of the substances used in the final part of the introduction, also underlining the novelties, while in the text (results section) we have tried to compare the results with what was obtained in the literature for other similar systems.

Minor remarks.

Please explain the abbreviation "GA", presumably on the line 132.

Some symbols are not defined in Eqs. (2) and (3).

Figure 2b does not contain any useful information and has to be excluded. It is sufficient to note that the phase angle does depend neither the glycerol content nor the frequency.

In accordance with what was suggested, we have inserted what was requested and eliminated the figure
